# Automated Training Data Generation from Spectral Indexes for Mapping Surface Water Extent with Sentinel-2 Satellite Imagery at 10 m and 20 m Resolutions

**Kristofer Lasko ***, **Megan C. Maloney**, **Sarah J. Becker**, **Andrew W. H. Griffin**, **Susan L. Lyon** and **Sean P. Griffin**

Geospatial Research Laboratory, Engineer Research and Development Center, U.S. Army Corps of Engineers, Alexandria, VA 22315, USA; megan.c.maloney@erdc.dren.mil (M.C.M.); Sarah.J.Becker@erdc.dren.mil (S.J.B.); Andrew.W.Griffin@erdc.dren.mil (A.W.H.G.); Susan.L.Lyon@erdc.dren.mil (S.L.L.); Sean.P.Griffin@erdc.dren.mil (S.P.G.)
* Correspondence: Kristofer.D.Lasko@erdc.dren.mil

**Abstract:** This study presents an automated methodology to generate training data for surface water mapping from a single Sentinel-2 granule at 10 m (4 band, VIS/NIR) or 20 m (9 band, VIS/NIR/SWIR) resolution without the need for ancillary training data layers. The 20 m method incorporates an ensemble of three spectral indexes with optimal band thresholds, whereas the 10 m method achieves similar results using fewer bands and a single spectral index. A spectrally balanced and randomly generated set of training data based on the index values and optimal thresholds is used to fit machine learning classifiers. Statistical validation compares the 20 m ensemble-only method to the 20 m ensemble method with a random forest classifier. Results show the 20 m ensemble-only method had an overall accuracy of 89.5% (±1.7%), whereas the ensemble method combined with the random forest classifier performed better, with a ~4.8% higher overall accuracy: 20 m method (94.3% (±1.3%)) with optimal spectral index and SWIR thresholds of −0.03 and 800, respectively, and 10 m method (93.4% (±1.5%)) with optimal spectral index and NIR thresholds of −0.01 and 800, respectively. Comparison of other supervised classifiers trained automatically with the framework typically resulted in less than 1% accuracy improvement compared with the random forest, suggesting that training data quality is more important than classifier type. This straightforward framework enables accurate surface water classification across diverse geographies, making it ideal for development into a decision support tool for water resource managers.

**Keywords:** surface water; water index; band ratios; machine learning; random forest; multispectral; automatic; MNDWI; AWEI; SCL

## 1. Introduction

Surface water is one of the most important natural resources on earth due to its life sustaining properties and importance for agriculture, biodiversity, and human society. Mapping surface water provides essential support for managing this important resource. Surface water extent is temporally dynamic, impacted by natural and anthropogenic drivers such as precipitation, evaporation, irrigation, dams, flooding, and glacier melt. Furthermore, of the total global inland surface water extent detected by Landsat between 1999–2018, only 60% was persistent, while the remaining 40% was ephemeral [1]. In the year 2000, inland surface water extent was unevenly distributed across the world, with over 70% located in North America and Asia, and less than 10% each in Europe, Africa, South America, and Oceania [2]. Many freshwater reservoirs, lakes, and other inundated areas have been experiencing declines in volume and surface water extent over recent decades across various parts of the world [3,4], with exception to some areas experiencing increases, such as the Tibetan Plateau [5,6]. Increases in loss of surface water extent have been found

in parts of the Middle East and Central Asia due to drought, damming, or unsustainable water withdrawal [7,8]. While recent studies identified some of the most concentrated losses in these regions, net gains in surface water extent have also been recorded across all continents except Oceania, largely attributed to reservoir filling and glacier melt from climate change [5,9,10] Changes in surface water extent, especially losses, impact land use, water resources management, and ecosystem services. Therefore, mapping support must be constantly updated to reflect changing conditions and support time-sensitive management decisions. Automated remote sensing of surface water is ideal for this purpose.

In order to obtain such surface water extent metrics, increasingly robust algorithms have been developed for mapping surface water extent with coarse resolution sensors such as the Moderate Resolution Imaging Spectroradiometer (MODIS) [11–13], including studies with daily temporal resolution [14]. More recently, studies show that moderate resolution sensors, such as Landsat 8, are able to capture change dynamics of smaller waterbodies.

Landsat-scale water mapping typically applies supervised classification methods that derive information from the spectral bands, a combination of well-documented spectral indexes such as the Normalized Difference Water Index (NDWI) or Modified Normalized Difference Water Index (MNDWI), and digital elevation models (DEM) [1,2,5,15–22]. More robust indexes such as the Automated Water Extraction Index (AWEI) were developed to improve classification accuracy in shadow and dark surface areas that are often missed by the MNDWI [23]. Additional work has been conducted to continually refine the spectral indexes for higher accuracy in challenging terrain with shadows [24,25]. More recently, some studies have leveraged deep learning to provide high accuracy output in complex environments [26–28]. The global availability of 10 m and 20 m Sentinel-2 imagery has enabled higher spatial and temporal resolution mapping especially within the challenging heterogeneous and urban landscapes [29–31]. Synthetic Aperture Radar is also used in conjunction with optical imagery to overcome issues with cloud cover and provide robust areal estimates [32–37].

While many surface water mapping studies use machine learning to provide more accurate results than with spectral indexes alone, training data collection remains a tedious and time-consuming step, or studies may rely on ancillary spatial layers to generate training data which may propagate errors due to inaccuracy. These layers, such as OpenStreetMap, often have temporal mismatches with the imagery and may omit small bodies of water not located near major cities. One recent study used Sentinel-2 imagery overlaid with OpenStreetMap to automatically extract water pixels [38]. A subsequent study used fuzzy membership functions, spectral indexes, and color transformations for automated training data generation [30], whereas unsupervised multidimensional hierarchical clustering with spectral indexes and individual bands were used to automate classification in France with a kappa score just below 0.9 [39]. Another study generated automated training and mapping with Sentinel-2 and Landsat 8 using land–water histograms with multitemporal imagery and five established water indexes [40]. The automated training data proved to be accurate in a limited, local study area with overall low root mean square errors in comparison to areal estimates. Another unsupervised method using mean-shift segmentation and a spectral index histogram to automatically generate training data found high accuracy in a study site of Southern Spain with an overall accuracy of 97%, and water user and producer accuracies of 90.2% and 88.9% [41]. While these methods prove useful and applicable across limited regions, studies have not yet developed accurate training data generation and automated mapping methods at the 10 m and 20 m spatial scales for Sentinel-2 imagery applicable to wide geographic areas, and more importantly, without the need for ancillary training layers (e.g., OpenStreetMap) to generate the training dataset and automate the workflow.

In this study, we develop two similar methods using 20 m and 10 m imagery to automate training data generation for mapping surface water extent. The 20 m method uses an ensemble of three reliable spectral indexes (NDWI, MNDWI, and AWEIsh) and optimal band thresholds in the SWIR spectrum to exclude false positive water pixels from the training data. Due to lack of SWIR bands, the 10 m method generates training

data using the NDWI with optimal band thresholds in the NIR spectrum. The training data is then applied to a machine learning classifier to map surface water extent at the Sentinel-2 granule level. On-demand generation of surface water extent for any given date or location is possible by only requiring a single Sentinel-2 granule for training and prediction. Additional information and details are discussed in the methodology section.

## 2. Datasets and Methods

### 2.1. Objective

While supervised classification is one of the most commonly used methods for mapping surface water extent, it requires manual selection and labeling of training data, which can be costly and time-consuming. Therefore, the main objective of this study was to develop a simple, efficient, and accurate algorithm for automatically generating surface water maps with high accuracy across a wide geographic range and in challenging environments such as urban areas and mountainous terrain using Sentinel-2 surface reflectance imagery at a spatial resolution of 10 m or 20 m. The algorithm should be self-contained without a need for ancillary spatial data layers for training, require no manual training data collection, and be applicable for automatically training and predicting on individual Sentinel-2 scenes containing surface water. This independent design would facilitate easy integration into a decision support tool for water resource managers.

### 2.2. Sentinel-2 Imagery (10 m and 20 m)

The Sentinel-2 Multispectral Instrument (MSI) imagery is provided by the European Space Agency [42]. We acquired the Bottom of Atmosphere (BoA) reflectance product, Level 2A (L2A), directly from the Copernicus Open Access Hub. The L2A product contains geometrically- and radiometrically-corrected imagery that is ready for analysis. It also includes a quality assurance layer that contains information on clouds, cloud shadows, snow, and ice, which we use to mask the input imagery. Sentinel-2 MSI provides a total of 13 spectral bands at three spatial resolutions (10 m, 20 m, and 60m). The 10 m L2A dataset contains 4 bands: red, green, blue, and near-infrared. In addition to those bands, the 20 m L2A product also contains three vegetation red-edge bands and two shortwave infrared bands [43]. These additional 20 m bands enable a wider array of spectral indexes for surface water mapping.

### 2.3. SRTM Digital Elevation Model

The Shuttle Radar Topography Mission (SRTM) DEM was acquired for the mountainous scene over Waterton Park, Canada (Sentinel-2 granule: T11UQQ) for error reduction in an optional post-processing step. The SRTM DEM is provided at 1 Arc-second (30 m) with void-filled data at the near-global scale. The dataset is freely available from the USGS EarthExplorer website. The SRTM dataset was acquired in February of 2000 onboard the shuttle Endeavour and it is the best available near-global DEM at 30 m spatial resolution. However, as with most DEMs, the SRTM DEM suffers some artifacts due to terrain shadow, vegetation canopy, or other dense obstructions [44].

In order to maintain the original resolution of the Sentinel-2 imagery and subsequently created surface water maps, the SRTM DEM was downscaled to 20 m and 10 m resolution. To reduce any potential spatial issues due to downscaling from 30 m resolution, we used a neighborhood window to smooth the downscaled pixels to better represent the underlying terrain. The downscaled SRTM DEM was then used to remove false positive water pixels on steep slopes with shadows at the T11UQQ site.

### 2.4. Software and Programs

For analysis of the satellite datasets we used Esri ArcGIS Pro version 2.5.2. Specifically, ArcPy functions within Python 3.6.7 were used in conjunction with GDAL, NumPy, and Scikit-Learn for image clipping, calculating spectral indexes, creating an ensemble of the three indexes, generating the equalized random points for training, labeling the training



points, training and applying the machine learning classifiers, and calculating accuracy metrics in batch. Specifically, the ArcPy Spatial Analyst functions were leveraged for much of the analysis. The ensemble of spectral indexes were created using the "CellStatistics" function, the equalized random points for automatic training were generated using the "CreateAccuracyAssessmentPoints" function with the NDWI layer as the stratifying layer, and "UpdateCursor" was used for iterating the points and applying SQL statements on the index ensemble and SWIR/NIR bands. Scikit-learn was used for the machine learning classifier training and prediction. Accuracy metrics were compiled using a combination of python Pandas library, csv library, and ArcPy functions such as "SearchCursor", and "ExtractMultiValuesToPoints". Python libraries MatPlotLib and Seaborn were primarily used for graphics and visualization [45].

*2.5. Study Sites*

For the analysis, we selected 12 study sites of approximately 5000 km² across diverse geographies that represent different water types. The sites were selected in order to capture spectrally diverse types of surface water features, including factors such as turbidity levels, flow rate, and presence of land cover which could contain confusing non-target spectra (e.g., impervious surfaces). These included: glaciated water features, clear water, moderately turbid water, highly turbid/sedimented water, coastal water, and agricultural water features. The selected study sites are shown in Table 1, where they are labeled with the corresponding Sentinel-2 granule number. The sites include: Lake Tahoe, USA (T10SGJ), Waterton Park, Canada (T11UQQ), Mississippi River, USA (T16SBG), Au Sable River and Lake Michigan, USA (T16TGQ), Indian River Lagoon, USA (T17RNM), The Bahamas (T17RRH), New River, USA (T17SNB), Madeira River, Brazil (T20LLR), Parana River, Paraguay (T21JWK), Amsterdam, Netherlands (T31UFU), Sundarbans spanning both Bangladesh and India (T45QYE), Hanoi, and the Red River Delta, Vietnam (T48QWJ). Overall, these study sites represent diverse and challenging environments to map with locations in Asia, Europe, North America, and South America. Since urban environments are one of the most difficult terrains in which to map surface water due to spectrally similar non-target shadows and infrastructure [46], 11 of the study sites include cities or small urban areas. The remaining site includes steep topography and contains shadows, snow, and ice, which are often falsely classified as water pixels. Moreover, the study sites include reservoirs, varying sizes of rivers, canals, wetlands, inundated agriculture (e.g., rice fields), etc. in order to include a variety of spectrally and spatially complex water features that are difficult to accurately map. Figure 1 contains a representative subset view of each study area.

**Table 1.** Overview and description of the study sites and associated Sentinel-2 imagery.

| Sentinel-2 Granule | Location | Date | Site Description | Water Type |
|:---:|:---:|:---:|:---:|:---:|
| T10SGJ | Lake Tahoe (USA) | 1 October 2020 | Suburban, rivers, mountain, snow | Clear |
| T11UQQ | Waterton Park (Canada) | 3 October 2020 | Glaciers, lakes, streams | Clear, glacier fed |
| T16SBG | Mississippi & Ohio River (USA) | 26 December 2020 | Meandering river | Turbid |
| T16TGQ | Au Sable River (USA) | 8 November 2020 | Great Lakes, urban, river | Turbid, coastal |
| T17RNM | Indian River Lagoon (USA) | 9 December 2020 | Wetlands, urban, beach, ocean | Coastal |
| T17RRH | Nassau & Andros Isle (The Bahamas) | 8 January 2020 | Ocean, urban, tidal | Coastal |
| T17SNB | New River (USA) | 7 November 2020 | Hilly, meandering river | Turbid |

**Table 1.** *Cont.*

| Sentinel-2 Granule | Location | Date | Site Description | Water Type |
|---|---|---|---|---|
| T20LLR | Madeira River (Brazil) | 11 August 2020 | Highly turbid river | Sedimented |
| T21JWK | Parana River (Paraguay) | 21 August 2020 | Highly turbid river | Sedimented |
| T31UFU | Amsterdam (Netherlands) | 15 April 2020 | Dense urban–water interface | Clear |
| T45QYE | Sundarbans (Bangladesh, India) | 27 December 2020 | Inland water, streams, turbid water | Sedimented, agriculture |
| T48QWJ | Hanoi (Vietnam) | 14 November 2020 | Urban, rice, turbid water | Turbid, agriculture |

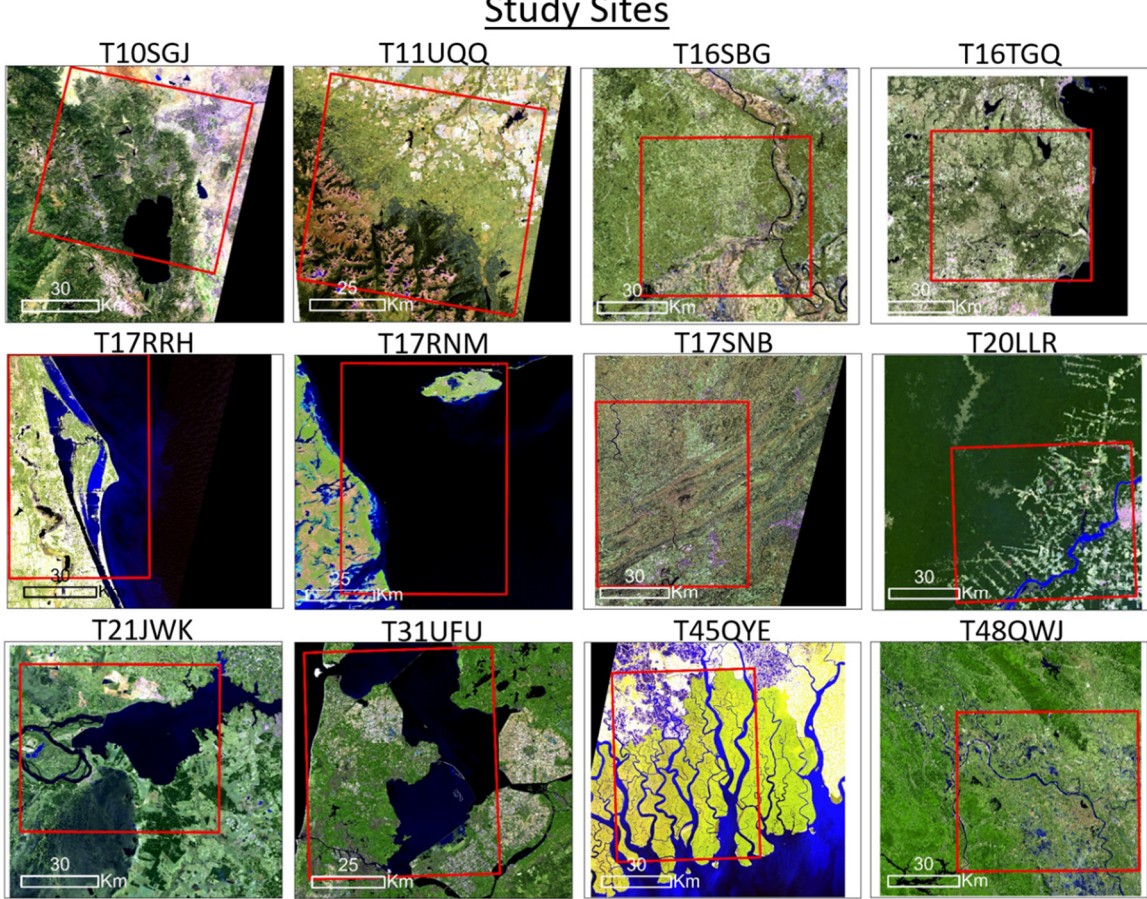

**Figure 1.** Overview of the twelve study sites shown in red outline. Each granule is visualized with Red, Green, Blue (RGB) channel bands of: SWIR1, Near-IR, and Red with varied histogram stretches and slightly varied zoom levels. Each study area is approximately 5000 km$^2$.

### 2.6. 20 m Framework: Automated Training Data Generation

This method was designed to automatically obtain surface water extent from cloud-masked or cloud-composited Sentinel-2 20 m BoA imagery as shown in Figure 2. The analysis was performed at the granule level. A granule is a single Sentinel-2 scene which is identified by a six-digit alphanumeric code as shown in Figure 1. For each granule, three established water indexes are calculated: Automated NDWI, MNDWI, and AWEIsh [23,47,48]. A starting threshold of 0 was preliminarily applied to each spectral index to separate surface water and non-water pixels, because this value typically balances omission and commission errors as suggested in the literature [48]. This step provided preliminary surface water extent maps for generating the ground-truth points used to

calculate optimal thresholds for the final surface water maps. Next, an index ensemble was created by combining the three indexes into a single raster with values ranging from 0–3, with 3 representing all indexes were above the threshold for a given pixel, 2 representing two indexes above the threshold, 1 for one index above the threshold, and 0 for no indexes above the threshold. Meanwhile, the NDWI raster was reclassified into six quantiles (3 above zero and 3 below zero) to ensure balanced sampling of land surface features along the spectrum of potential index values. Then, 8000 points were generated within the six quantiles of the reclassified NDWI raster using equalized random sampling. This quantity was necessary in order to minimize noise from erroneous training points. Lastly, values from the spectral bands and spectral indexes were extracted for each point.

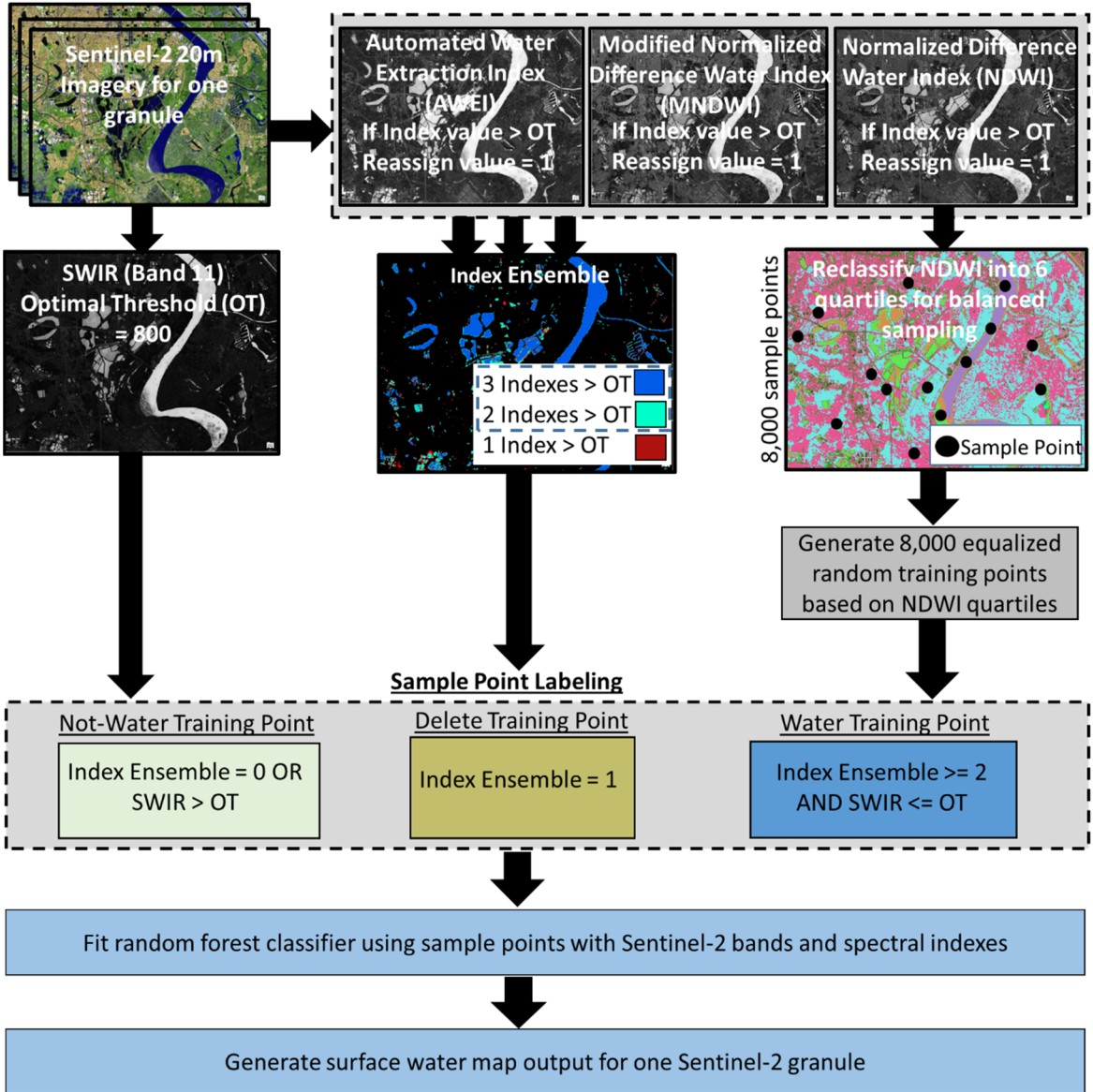

**Figure 2.** Methodology for 20 m Sentinel-2 automated training data generation and initial surface water extent map generation. The preliminary set of maps are generated with a standard zero threshold for the indexes and 850 BoA reflectance threshold for SWIR1 band. Ground-truth points are generated on this initial set of maps and then used to determine the Optimal Thresholds (OT) for the spectral indexes and SWIR band. The OTs are used to create the final surface water maps.

The training dataset was developed by separating the 8000 training points into 'surface water' and 'not surface water' points. The following were used to determine class

assignment for a given point: if 0 indexes were above the threshold, the point was marked as 'not water'; if 2 or 3 of the indexes were above the threshold, the point was marked as 'surface water'; and if only 1 index was above the threshold, the point was deleted, as it is uncertain as to which class it belongs. Additionally, the shortwave infrared (Band 11) spectrum is highly sensitive to water and was used with a threshold to reduce false positive water pixels [49], as urban pixels often exhibit index values similar to water [50]. Training data points with SWIR1 BoA reflectance values greater than 850 were marked as 'not water'. The final training dataset was input into a random forest classifier to predict separate water/not water outputs for each Sentinel-2 granule/scene [51]. We note that the training dataset was generated independently for each scene.

### 2.7. Creation of Ground-Truth Points to Calculate Optimal Thresholds

After generating the initial set of surface water maps for the twelve sites, the ground-truth points were generated. Ground-truth points were generated on the initial NDWI threshold maps from the previous section. These points will be used for computing the optimal thresholds for the spectral index and band thresholds in subsequent analysis. Based on binomial probability, a total of 2400 points were required, spread evenly across the 12 locations to obtain a 2% margin of error and 95% confidence level [52]. While stratified random sampling is the standard for most land cover-based accuracy assessments [53], this method would be unsuitable for binary mapping analysis as it would result in an insufficient number of water points generated due to the low proportion of water across all scenes (average = 22%). Instead, this study used an equalized random sample of 100 water and 100 not water points per site (2400 points per method) separately for both the 10 m and 20 m datasets (4800 points total). The sample was generated approximately equally across the six reclassified NDWI quantiles to establish a balanced range of spectral values for evaluation. Since the points determine the optimal thresholds, we generated additional points that target the most error-prone regions of the map. Therefore, an additional 20 points for each scene (240 total, 120 water, 120 not water) were generated for areas prone to higher error, identified as an area where there are no more than two contiguous pixels of the same class (surface water or non-surface water) since water features tend to be clustered in large continuous groupings (e.g., rivers, lakes, etc.). The ground-truth was interpreted for each point based on expert assessment using high-resolution imagery (Worldview-2, pan-sharpened with nominal resolution of 0.5m) in ArcGIS Pro, and the original Sentinel-2 imagery [52].

### 2.8. Determination of Optimal Thresholds for Spectral Indexes and Bands

After the first set of intermediate water maps was created using the initial zero threshold for each spectral index, and the associated ground-truth points had been labeled, we determined the optimal threshold for the indexes. For the spectral indexes we tested threshold values ranging from $-0.25$ to $0.25$ at an interval of $0.01$, which were considered to be the possible candidate values. We then generated the water maps for each of the values using the same methods described in Figure 2. The optimal threshold was selected by evaluating the overall accuracy with respect to the 2400 ground-truth points.

After the optimal index threshold was determined, we calculated the optimal SWIR band threshold by iterating the framework separately for each scene using possible SWIR BoA reflectance thresholds ranging from 50 to 2000 at intervals of 50. The purpose of the SWIR threshold was to remove false positive water pixels, especially within highly reflective urban pixels. The outputs for all scenes and all thresholds were compared to the ground-truth points using a confusion matrix to determine the optimal threshold.

### 2.9. Sentinel-2 10 m Method for Automated Training Data Generation

Unlike the 20 m Sentinel-2 L2A product, the 10 m product does not contain red-edge or shortwave infrared bands, which severely restricts the possible water indexes that can be leveraged. In this case, only the NDWI was used. The 10 m method follows the same

processing and analysis framework as the 20 m, with several small exceptions. Because only the NDWI was used, an index ensemble cannot be generated and the automation of the training data selection was instead based on two conditions for a given pixel: (1) NDWI > threshold = 'water', NDWI < threshold = 'not water', and (2) near-infrared band > threshold = 'not water'. The initial threshold of 0 was used for NDWI and 900 for near-infrared BoA reflectance in order to generate the 2400 ground-truth points used for calculation of optimal thresholds. Similar to the SWIR, the NIR threshold helped to exclude false positive urban pixels from the training data. As with the 20 m dataset, once the framework was run with the initial values and the initial surface water extent maps were generated, 2400 new equalized random ground-truth points (1200 surface water, 1200 not surface water) were then generated and interpreted. The optimal NDWI and NIR BoA reflectance thresholds were then determined by iterating the framework with thresholds between −0.25 to 0.25 (NDWI) and 0 to 2000 (NIR) respectively, generating surface water extent maps at each value. The accuracy of each iteration was calculated from a confusion matrix using the ground-truth points.

### 2.10. Optional Post-Processing with SRTM DEM

The SRTM 30m DEM was used to remove false positive water pixels from extremely steep slopes in mountainous terrain. False positive water pixels commonly occur in steep, sloping terrain due to the inherently low spectral values when the terrain is obscured from direct sunlight and thus in a shadow. While it is often standard to include a DEM as an input for classification, we instead used it as an optional post-processing step so that this study can be less data intensive, as it is intended for time-sensitive water mapping applications by end-users. The mountainous scene over Waterton Park, Canada (T11UQQ) was chosen to test this post-processing method due to the prevalence of shadows in the steep terrain. We calculated slope for this location, resampled the SRTM DEM to Sentinel-2 resolution, then smoothed the result with a 3 × 3 neighborhood pixel window to prevent removal of features like steep riverbanks and account for the mismatch in spatial resolution. Finally, we applied various slope thresholds to mask out false water pixels. We tested and compared overall accuracy for thresholds ranging from 0 to 30 degrees at 0.5 degree intervals in order to determine the optimal threshold.

### 2.11. Accuracy Asssessment

We used the 4800 ground truth points described in Section 2.7 (2400 for the 10 m, 2400 for the 20 m) for evaluating the accuracy of the automated training data generation frameworks following good practices in accuracy assessment [52]. We used the weighting from the mapped proportions of each of the two classes and the confusion matrixes of pixel counts to calculate the unbiased areal estimates for each class [53]. This resulted in adjustments to the accuracy metrics based on the unbiased areal estimates created from the user's accuracies. We also computed 95% confidence intervals based on the same, which enables comparison among the different maps.

We compared the accuracy resulting from the 2400 points, as well as with the additional 200 error prone points. These error prone points, as described in the methods section, were generated to ensure robust analysis for testing the thresholds in challenging spectral conditions.

Lastly, we evaluated the Sentinel-2 Scene Classification Layer (SCL) which is provided with all L2A imagery [43]. The SCL is provided at 20 m and contains masks for specific layers such as snow, clouds, smoke, vegetation, and surface water calculated using spectral indexes and thresholds.

### 2.12. Comparison of Classifiers

Studies often debate about which supervised classifier provides the best performance for a given application. Some of the most common and robust supervised machine learning classifiers include the random forest [51], gradient boosted trees (GBT) [54], support vector

machine (SVM) [55], as well as a majority vote classifier based on these three classifiers. The random forest uses bootstrap aggregated sampling to build individual decision trees. Within the structure of a tree, a random sample of the square root of the number of predictors was chosen for each split as best candidates derived from the entire predictor set. For the SVM, the optimal hyperplane (dimension = n_features) was selected based on maximizing the distance between data points of the two classes. The GBT classifier operates similar to the random forest, but the individual trees are constructed in a series. These classifiers have different advantages and disadvantages with SVM being generally better at modeling linear dependencies and working with sparse data. However, while the GBT classifier also includes an ensemble of decision trees, it constructs each tree one at a time and uses each tree to improve the weaknesses of the next tree. The decision trees (GBT and random forest) can better handle non-linear dependencies and are faster to calculate than an SVM or majority vote classifier. The majority vote classifier combined all of the classifiers to leverage strengths of each one, but requires increased computation time.

We fed the automated training data into each of these classifiers with mostly default settings in scikit-learn (except n_estimators = 500, kernel = linear) and evaluated the accuracies using the ground-truth points. Because our training data was automatically created and applied individually for each granule without application to other datasets, model overfitting was not a significant concern. In order to quantify the improvement produced through the automated training data selection method with supervised classifiers, we also compared the outputs to the original ensemble of water indexes rasters (20 m only).

## 3. Results

### 3.1. Optimal Thresholds

We first calculated the optimal thresholds for the spectral indexes by evaluating accuracy with respect to the ground-truth points. In this first step, a SWIR threshold (20 m) or NIR threshold (10 m) was not included; however, we did apply a random forest classifier. Figure 3 shows the overall accuracy for each granule/scene, as well as the Sentinel-2 granule average accuracy based on changes to the spectral index thresholds. The 10 m dataset only includes NDWI and the 20 m dataset includes NDWI, MNDWI, and AWEIsh. Overall, the granule average indicated highest accuracy was achieved with a threshold of $-0.01$ for the 10 m dataset and $-0.03$ for the 20 m dataset. For the 20 m dataset, thresholds of $-0.02$, $-0.01$, and 0 were only 0.1% less accurate lower than the optimal threshold. Variability of the optimal threshold was observed across the different study sites. The two sites with a strong urban presence (T31UFU and T48QWJ) had thresholds slightly above 0 in the 20 m dataset, but thresholds of $-0.05$ and $-0.02$ respectively in the 10 m dataset. One of the granules dominated by shallow water and wetland areas had the highest threshold for the 20 m dataset (0.2), but not the 10 m dataset ($-0.01$) (T45QYE). We attribute some error due to the difficulty in generating ground-truth for this region within very shallow water agricultural pixels. Interestingly, there was no clear pattern between the granules/scenes based on water types. For example, T20LLR and T21JWK had the most turbid water, but the thresholds were not similar for both 10 m and 20 m datasets. The highest overall accuracy for the 10 m dataset without any NIR threshold was 84.9%, and for the 20 m dataset without any SWIR threshold was 87.5%.

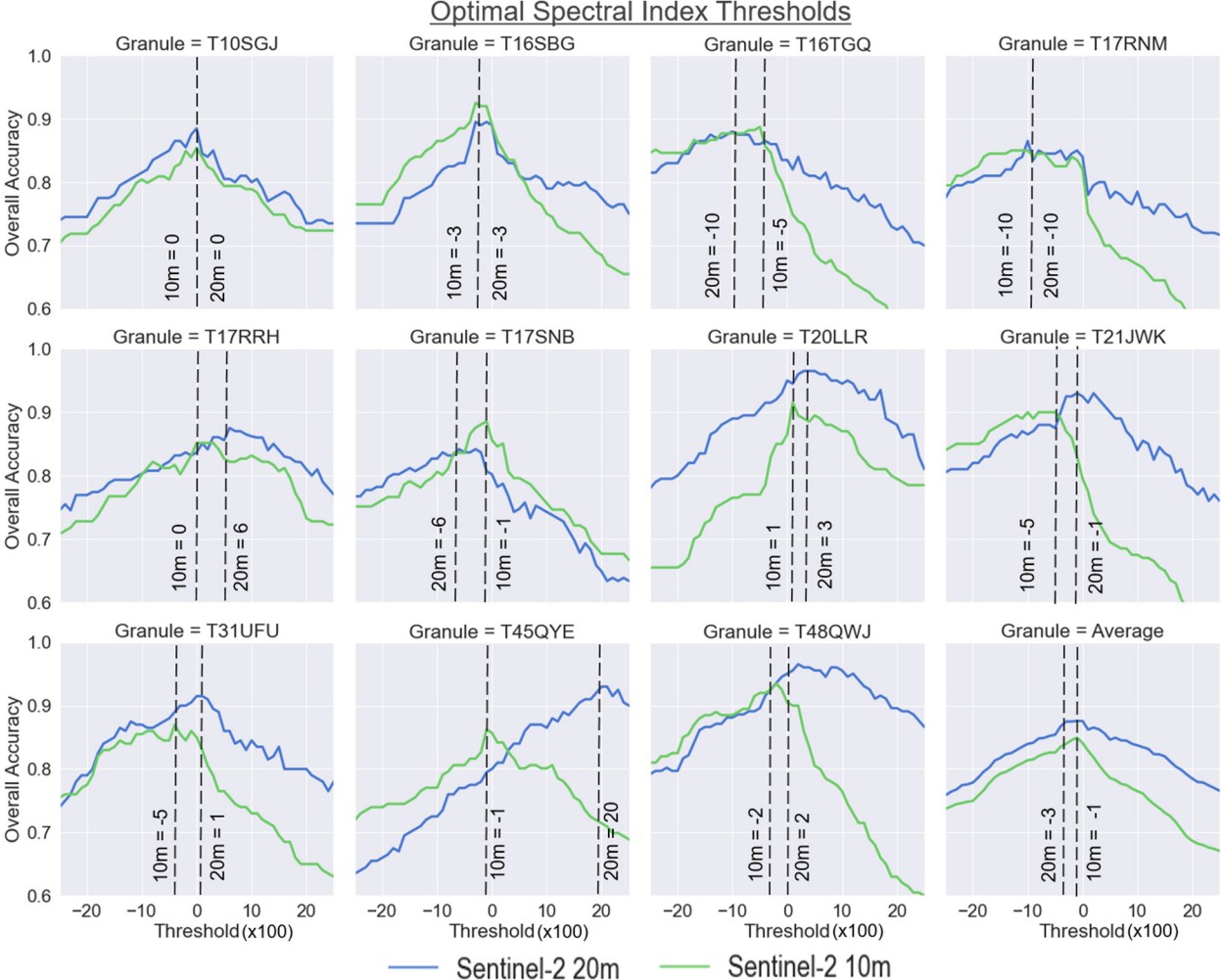

**Figure 3.** Evaluation of the most accurate spectral index threshold for automated training data selection for the 10 m and 20 m datasets across the test sites. For the 10 m dataset, the NDWI index is used, and for the 20 m dataset, NDWI, MNDWI, and AWEIsh are used. Overall highest accuracies (84.9% 10 m, 87.5% 20 m) are observed around –0.01 or 0 but vary by granule. Note: spectral index thresholds values are multiplied by 100 in the graph.

We subsequently evaluated the optimal thresholds for the SWIR (20 m) and NIR (10 m) datasets applied in conjunction with the optimal spectral index threshold and random forest classifier. The purpose of the SWIR and NIR band threshold was to remove false positive water pixels typically associated with urban and built-up areas. After evaluating the confusion matrices for the optimal spectral indexes, we tested the SWIR and NIR thresholds in conjunction with spectral index thresholds of −0.01 (10 m dataset) to −0.03 (20 m dataset) as determined from the previous section, and fed the training into a random forest classifier. Figure 4 shows the overall accuracy with respect to the NIR and SWIR thresholds starting at 50 and ending at 2000 with intervals of 50. Overall, when averaging across the twelve granules, the ideal NIR and SWIR thresholds were both 800. However, across the different scenes, the ideal thresholds varied between 550 and 1500 for SWIR20 m and 600 to 1550 for NIR10 m. Relatively similar thresholds and patterns of overall accuracy were observed between T16SBG, T16TGQ, T17RNM, T17RRH, T20LLR, and T21JWK. Overall, the accuracy improved significantly when using a NIR or SWIR threshold, compared to not using one. The accuracies with respect to the ground-truth are described in a subsequent section.

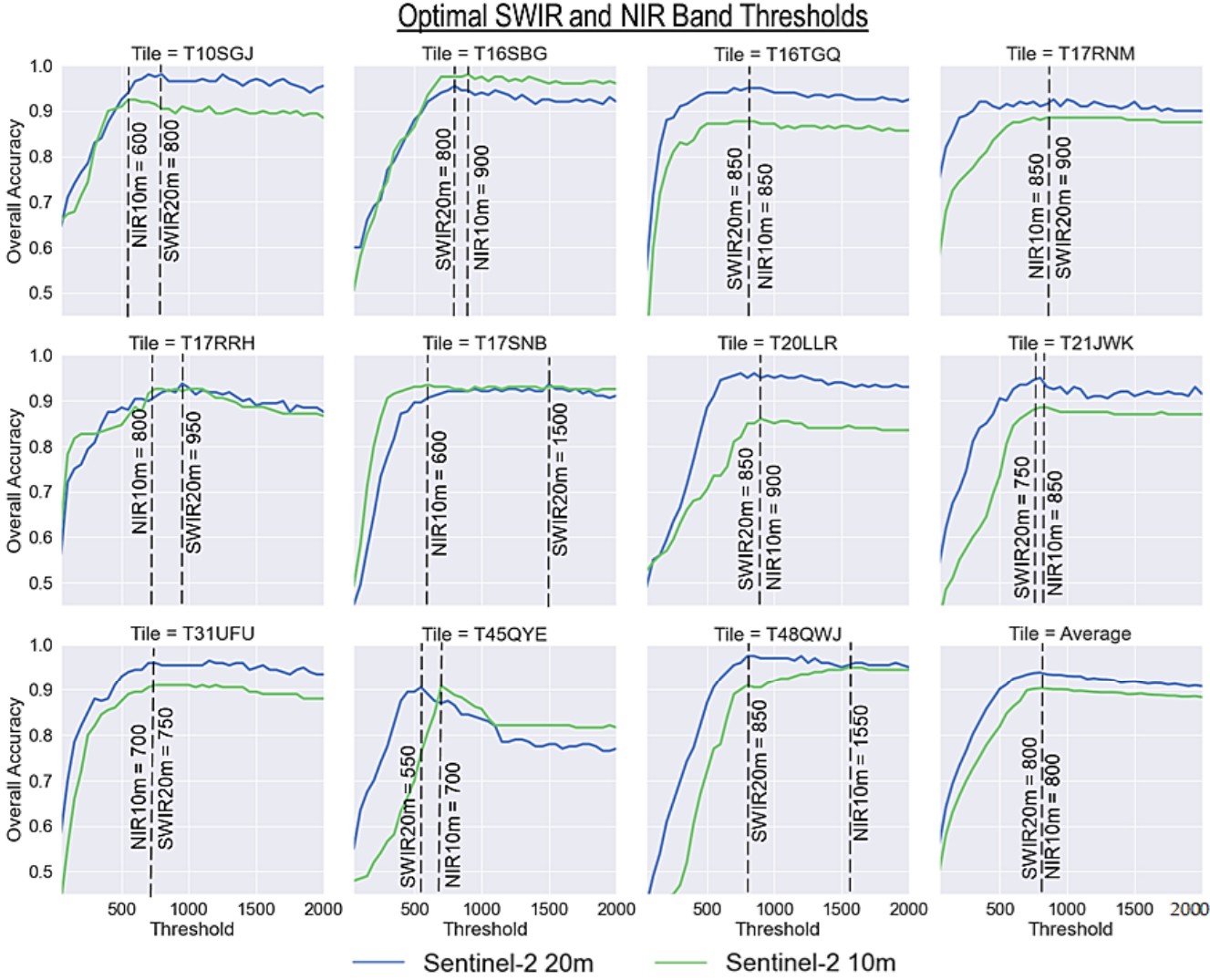

**Figure 4.** Evaluation of the most accurate band threshold for the 10 m and 20 m datasets across the test sites. For the 10 m dataset, the NIR band is used (due to lack of SWIR), and for 20 m dataset, the SWIR1 band is used. Overall accuracy improves significantly (>90%) when a band threshold is applied as compared with no threshold in the previous figure.

### 3.2. Removal of False Positive Water Pixels with SRTM DEM

To test the optional post-processing step, the SRTM 30 m DEM was converted into slope (degrees) and used to mask out steep sloping water pixels after the surface water map has been generated. The study site T11UQQ was the only one with considerable mountainous terrain and the DEM was only applied there. We evaluated the optimal slope threshold by calculating and comparing the overall accuracy with respect to the ground-truth points at slope thresholds ranging from 0 degrees to 30 degrees at intervals of 0.5 degrees. The graph of how the overall accuracy changes with the slope threshold is shown in Figure 5. Without a DEM correction the overall classification accuracy for this scene was only 75.1% for the 20 m Sentinel-2 framework. The maximum accuracy was achieved at 17 degrees slope with an overall accuracy of 96.6%, an improvement of over 20% without a DEM. A similar pattern was observed for the 10 m imagery.

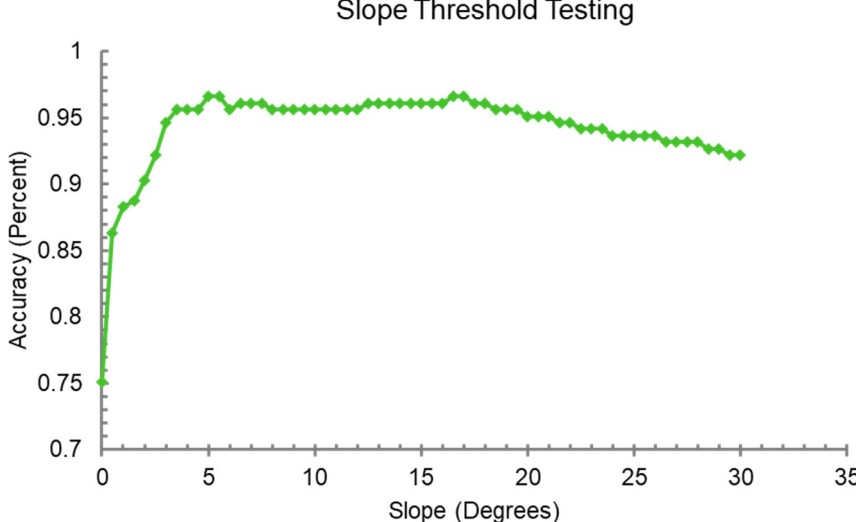

**Figure 5.** Slope calculated from a Digital Elevation Model (DEM) was applied to the mountainous granule T11UQQ. Pixels marked as water but with steep slopes were removed. This graph shows accuracy goes from 76% without a DEM, to as high as 96% with a slope degree threshold of about 17 degrees.

### 3.3. Sentinel-2 10 m and 20 m Surface Water Maps

Surface water maps were generated for both the 20 m Sentinel-2 framework and the 10 m Sentinel-2 framework using the calculated optimal thresholds from the previous sections in conjunction with a random forest classifier. The spectral index thresholds of −0.03 (20 m) and −0.01 (10 m) were used together with NIR (10 m) and SWIR (20 m) band thresholds of 800. As mentioned in the methods section, the training data points are automatically generated based on these criteria and the ensemble of indexes and then fed into a random forest classifier to produce the surface water maps. Figure 6 shows the results for each of the 12 study sites. At a small scale, the outputs for both the 20 m and 10 m products appear to do very well in comparison to the input imagery. The inundated agricultural fields and wetlands are captured in T48QWJ and T45QYE along with the turbid rivers. The small meandering rivers are well captured in T17SNB, and the coastal waters in T17RRH and T17RNM also are mapped well. In T11UQQ, glacier lakes are detected and ice-covered mountains and terrain shadow are largely and correctly omitted. Table 2 compares the surface water area mapped in the 10 m and 20 m products. Overall, they produced similar outputs with only an average difference of 4.3%. However, differences in excess of 10% were observed at T48QWJ and T20LLR, which are likely attributable to omission error of paddy rice fields in the 10 m product at T48QWJ, and omission of the highly turbid portion of the Madeira River in the 10 m product at T20LLR. Otherwise, 6/12 of the study sites had less than 1% difference in surface water area between the 10 m and 20 m maps.

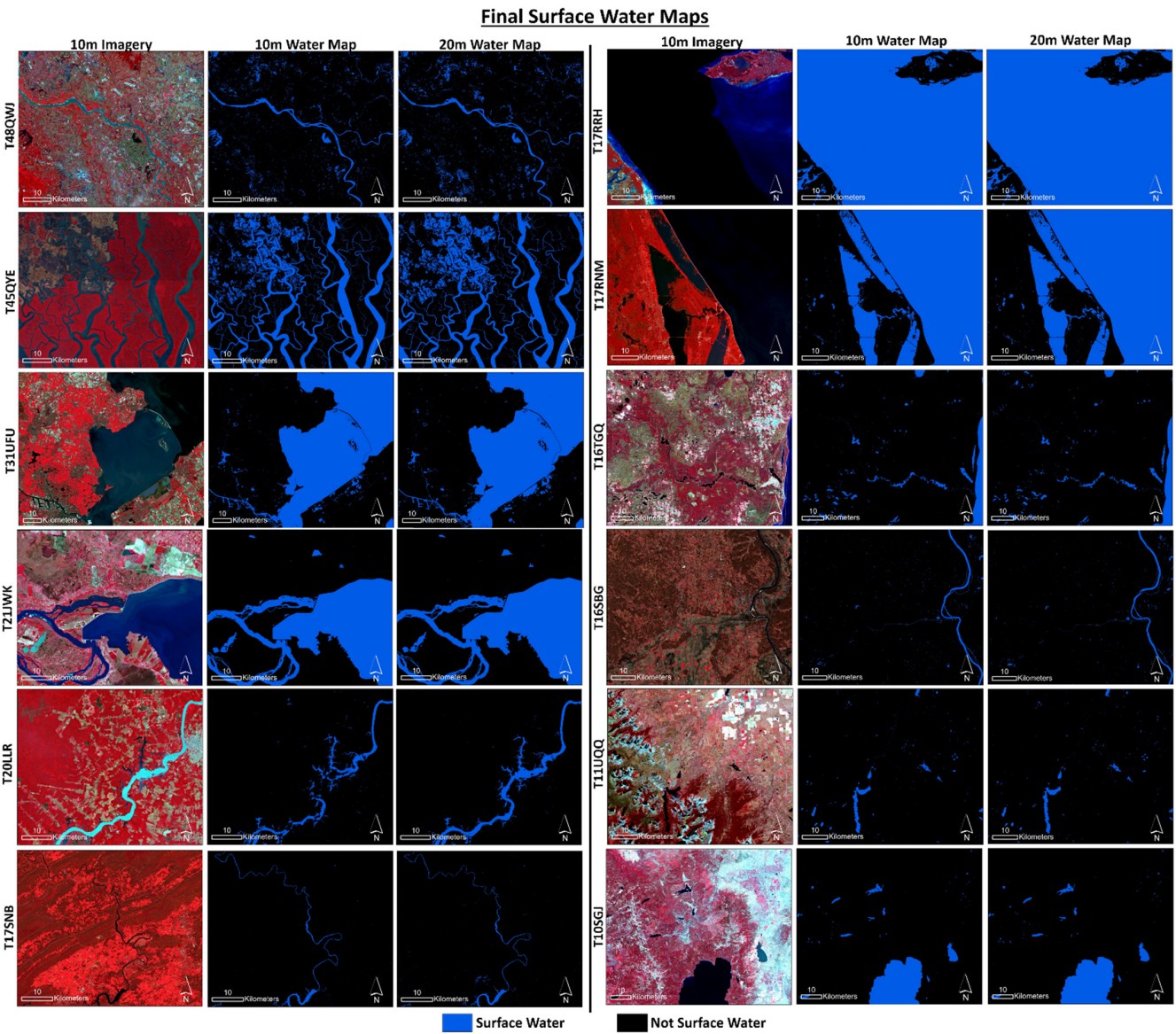

**Figure 6.** Comparison of the final 10 m and 20 m surface water maps across the 12 international test sites shown with the 10 m Sentinel-2 imagery in false color composite. The Sentinel-2 granule identifier is listed on the left side of the imagery.

**Table 2.** Comparison of the 10 m and 20 m surface water maps showed an average 4.3% difference in area.

| Granule | 10 m Water Area (km$^2$) | 20 m Water Area (km$^2$) | Percent Difference (%) |
|---|---|---|---|
| T10SGJ | 431 | 430 | 0.2 |
| T11UQQ | 117 | 116 | 1.3 |
| T16SBG | 98 | 92 | 6.1 |
| T16TGQ | 211 | 210 | 0.6 |
| T17RNM | 2748 | 2754 | 0.2 |
| T17RRH | 4608 | 4632 | 0.5 |
| T17SNB | 41 | 45 | 9.4 |
| T20LLR | 136 | 165 | 18.7 |
| T21JWK | 1357 | 1354 | 0.2 |
| T31UFU | 2324 | 2329 | 0.2 |
| T45QYE | 1482 | 1537 | 3.7 |
| T48QWJ | 288 | 320 | 10.4 |

### 3.4. Accuracy Assessment and Analysis of Errors

Figure 7 shows large-scale views of three of the study sites, highlighting some patterns of errors that were observed. Specifically, in T31UFU, within the city of Amsterdam, the 20 m product has relatively few buildings marked falsely as water, but the classification does not completely capture the canals and small waterways as designated in the figure. The 10 m product unsurprisingly does a better job at capturing the small canals, but still contains some omission error. This improved canal detection comes at the expense of more commission errors of buildings and shadows. The canal designated on both maps was difficult to map as it was also partially obstructed by tree cover. The 20 m SCL performed about the same as the 20 m product. In T48QWJ, close to the city of Hanoi, both the 20 m and 10 m products have minimal commission errors. However, omission errors are prevalent in the 20 m product along the narrow river and canals as marked in the figure. The 10 m product detects the canal, but omits some of the inundated paddy rice fields. The 20 m SCL water mask failed to detect much of the canals, aquaculture and paddy rice. In T20LLR, this portion of the Madeira River is highly turbid and was not detected in the 10 m product, but the 20 m product was able to clearly delineate it. A common thread among these scenes was that the omitted pixels often have spectral index values significantly below the optimal threshold. For example, in T20LLR, the NDWI across the omitted portion of the river in the 10 m output was highly negative and not indicative of water as it ranged from $-0.2$ to $-0.65$. In the same scene, the SCL failed to detect several small water bodies.

The SRTM DEM resulted in an improvement from 75% accuracy to 96% accuracy for granule T11UQQ. Figure 8 shows an example mountainous area for the 10 m and 20 m dataset where errors were present in the original result, but were mostly removed after the DEM was applied. The 20 m SCL water mask does not use a DEM, yet it appeared to have fewer shadow errors than the 20 m product from this study. However, after applying the DEM to the 20 m product from this study, it removed most of the shadow that remained in the SCL water mask.

Overall, the 20 m framework with random forest classifier had the highest accuracy at 94.3% ($\pm1.3\%$) with producer's accuracy of 91.5% and user's accuracy of 97.4% for surface water, and producer's accuracy of 97.4% and user's accuracy of 91.5% for the not water class. The 10 m framework with random forest classifier had an overall accuracy of 93.4% ($\pm1.5\%$) with producer's and user's accuracies of 90.6% and 90.5% for surface water, and 95.4% and 95.5% for the non-surface water class. We compared the 20 m ensemble index-only method with the ensemble index method and random forest classifier in order to quantify the improvement that the automated training data selection with random forest classification offers. The 20 m ensemble index-only method had an overall accuracy of 89.5% ($\pm1.7\%$). This was about 4.8% lower than the automated training method with random forest classifier, suggesting that the automated training method provides significant improvement over the standard spectral index ensemble approach.

As expected, the accuracies decrease when using the high error-prone additional points (AP). The overall accuracy for the 10 m dataset with AP was 91.2% ($\pm1.6\%$), and for the 20 m dataset the overall accuracy was 92.5% ($\pm1.5\%$). This suggests that our APs were effective at targeting challenging and difficult to map water features as we intended for optimal threshold selection.

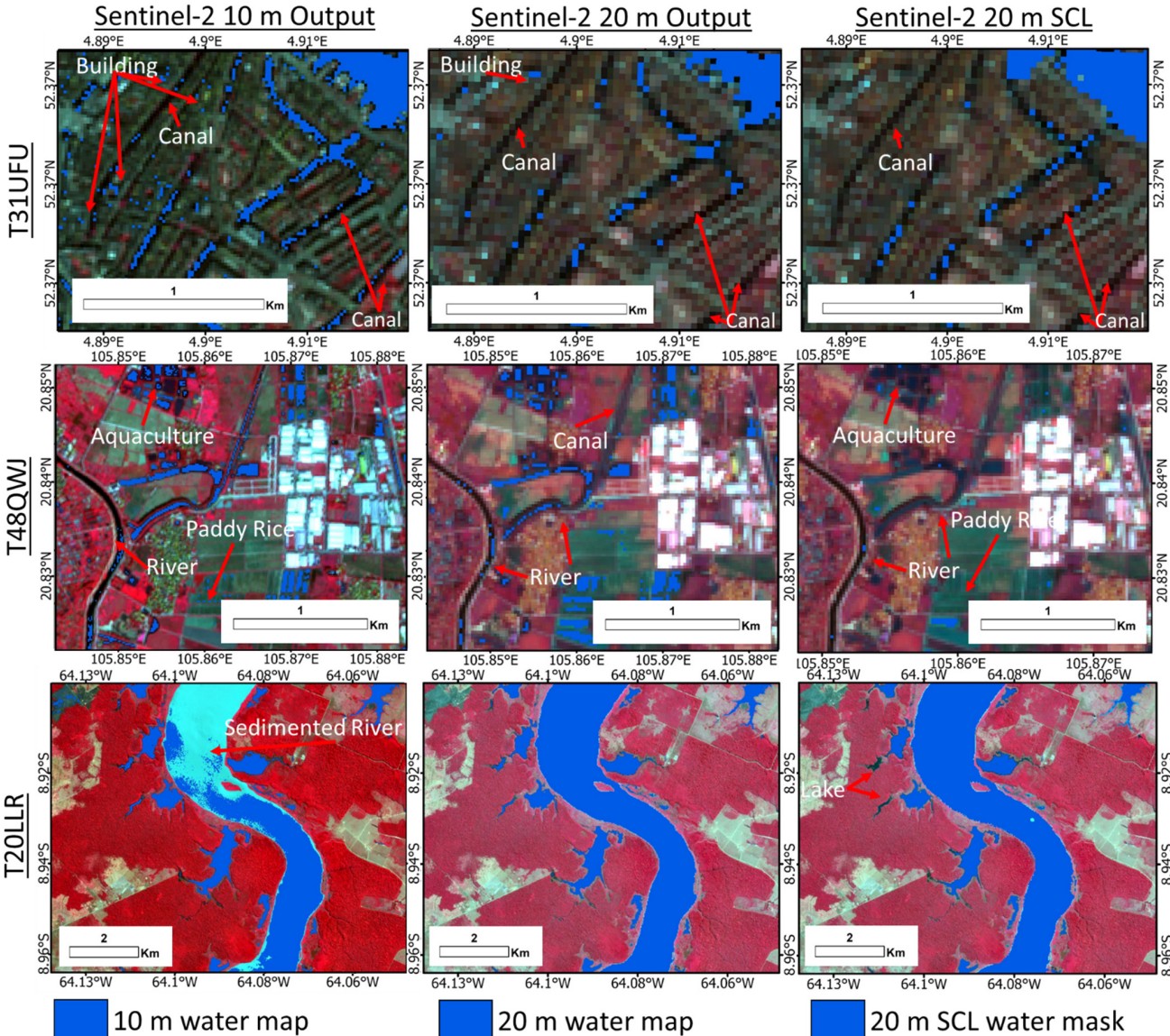

**Figure 7.** Example map errors in Amsterdam, Hanoi, and Madeira River. In Amsterdam, the 20 m product has fewer building commission errors, but has more omission errors of canals and small waterways. The 10 m layer captures more of the canals, but still has some omission errors. It also has more commission errors of buildings and shadows. The canal designated on both maps was difficult to capture as it was partially obstructed by tree cover.

The overall accuracies for each of the 12 study sites are shown in Figure 9. The comparison between the APs and standard accuracy assessment are also shown in this figure. For the 20 m framework, the lowest accuracy was found in T45QYE, likely due to the challenging shallow water pixels and water features obstructed by broadleaf forest in the mangrove swamps. Both the 10 m and 20 m frameworks had relatively low accuracies (88.6% and 91.7%) at the T16TGQ study site. This site contains several very narrow rivers that were not effectively captured by either dataset due to mixed water/land pixels that resulted in the water indexes producing highly negative values. Interestingly, the highest overall accuracy was observed at the T17RRH site for 10 m framework with 99.5%, while the highest overall accuracy for the 20 m framework was found in T31UFU with 96.7%.

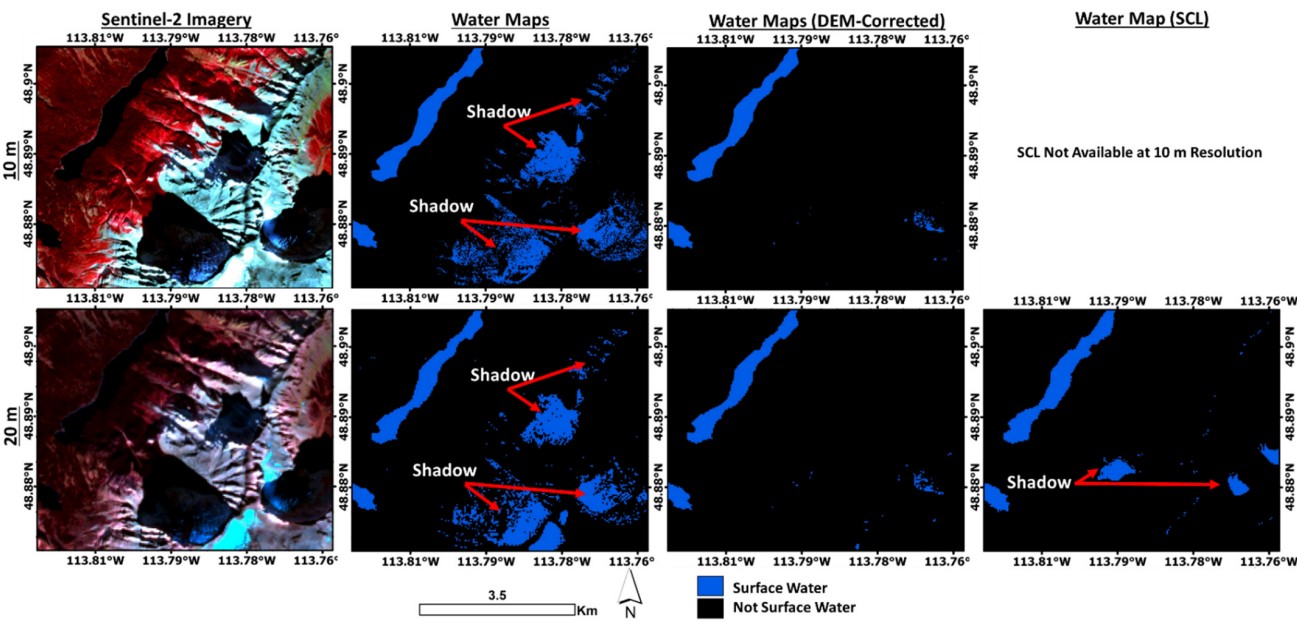

**Figure 8.** Mountainous area in the T11UQQ granule shown with the 10 m and 20 m imagery. Errors were present in the original output, but were significantly reduced after the DEM post-processing was applied.

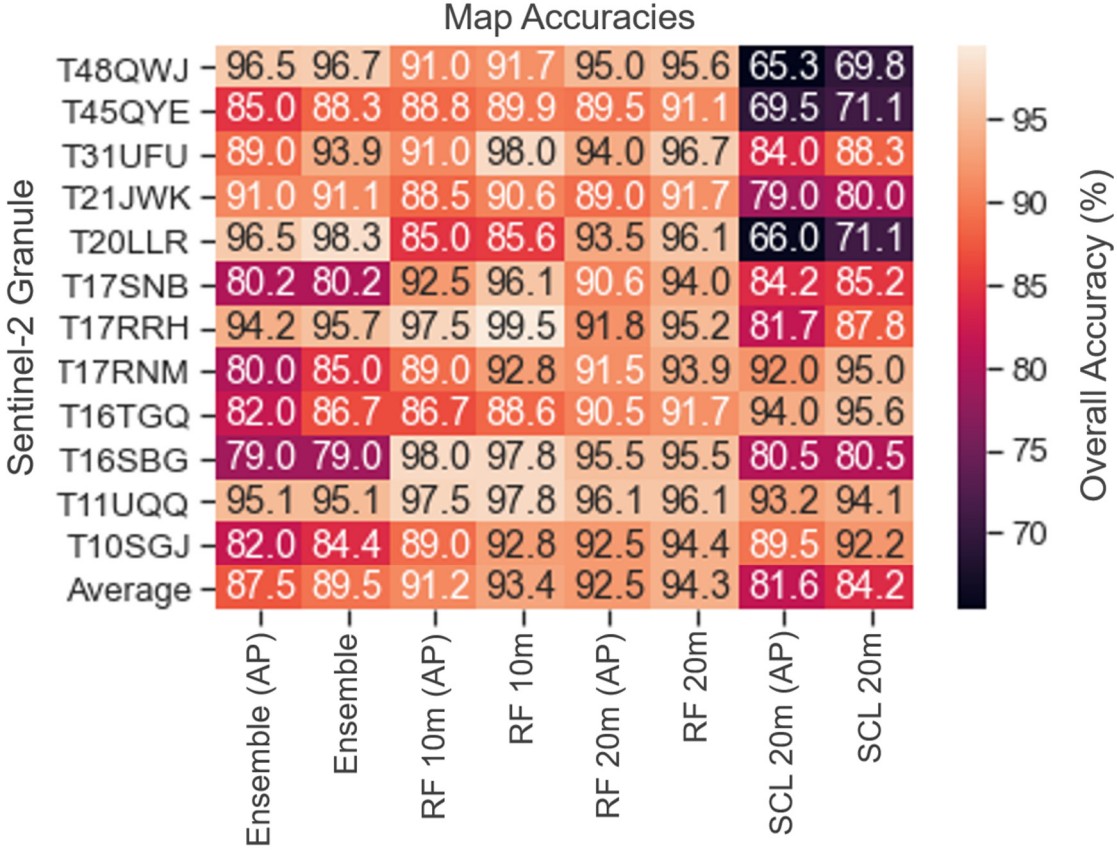

**Figure 9.** Overall accuracy for each Sentinel-2 granule and the average for the four different algorithms (ensemble-only, 20 m ensemble with random forest, 10 m index with random forest, and Sentinel-2 20 m SCL algorithm). The accuracies are compared for both a) 100 water and 100 not water points per site stratified in quantiles of NDWI, and b) with 20 additional points (AP) generated per granule on likely error pixels (isolated 1- or 2-pixel clusters).

The 20 m Sentinel-2 SCL water mask provided with all L2A imagery had an overall accuracy of only 84.2%, which was about 10.1% less accurate than the automated method with random forest from this study. It also had lower accuracy than the ensemble spectral index method from this study, which had 89.5% accuracy. Because of this accuracy, it is recommended for users to avoid relying on the SCL water mask, which performed especially poor in sites with high turbidity water (e.g., T20LLR Brazil: 71.1% accuracy) and with inundated agriculture (e.g., T48QWJ Hanoi: 69.8% accuracy, T45QYE Bangladesh: 71.1% accuracy).

Figure 10 shows the distribution of the errors based on the ground-truth points for each of the datasets. Overall, for both the 10 m and 20 m framework the false negative water pixels (omission errors) had a median NDWI of −0.045. Whereas the false positive water pixels (commission errors) for the 10 m dataset had a median of 0.014, the 20 m dataset had a median of −0.03, with a wider distribution of erroneous values. These distributions of errors could be used to target future improvements to the automation of the training data by focusing efforts on these NDWI values. We specifically chose NDWI for error analysis because while it may not be the most robust index, it was the common index among the 10 m and 20 m datasets due to the band limitations. The errors for the water mask from the 20 m SCL are also shown and interestingly the false negative NDWI values are much higher than the results from this study.

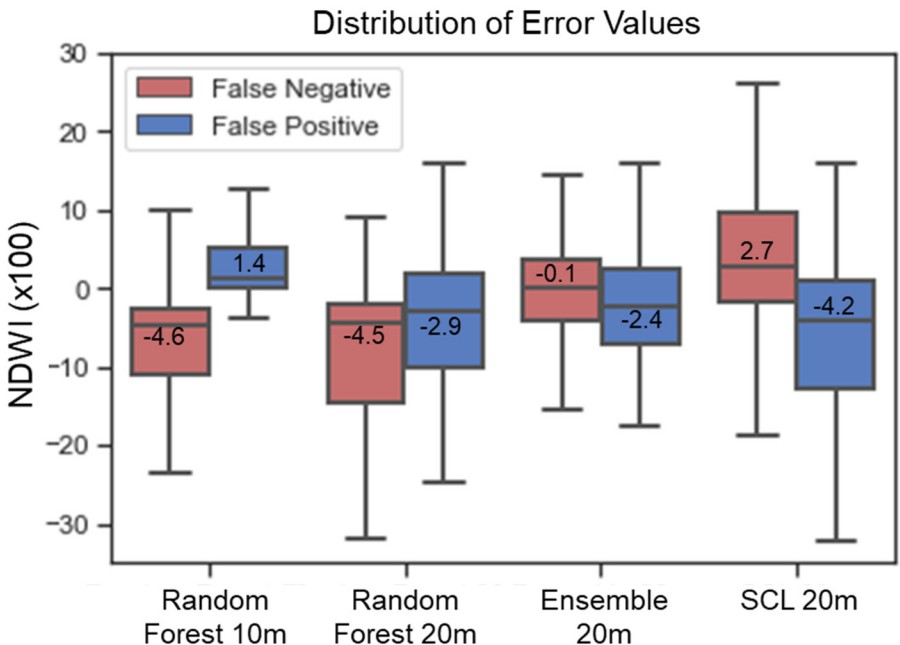

**Figure 10.** NDWI error patterns shown for each dataset including the index method with random forest (10 m), ensemble index method with random forest (20 m), ensemble only method (20 m), and the water mask from the scene classification layer (SCL 20 m).

*3.5. Comparison of Classifiers*

We modified the 10 m and 20 m frameworks to run random forest, gradient boosted trees, and support vector machine classifiers, as well as a majority vote classifier based on the combination of the three using the AP accuracy points for validation. The results for the 10 m and 20 m automated frameworks are shown in Table 3. Ultimately, the accuracies of the 10 m dataset were relatively similar across the different classifiers, with the highest for the SVM (91.5%) and the lowest for the GBT (91.1%). Whereas, for the 20 m framework the majority vote classifier had 92.6% accuracy, but was only 0.1% more accurate than just the random forest classifier alone. The SVM in this case had the lowest accuracy at 91.6%.

**Table 3.** Evaluation of classifiers using the automatically generated training data with high error prone ground-truth points (APs). Values are shown as a percentage (%).

| | Overall Accuracy | Water, User's Accuracy | Water, Producer's Accuracy | Not Water, User's Accuracy | Not Water, Producer's Accuracy |
|---|---|---|---|---|---|
| **20 m Accuracy with Additional Points (APs)** | | | | | |
| Majority Vote Classifier | 92.6 | 92.9 | 93.7 | 91.3 | 92.3 |
| Random Forest | 92.5 | 92.7 | 93.6 | 91.1 | 92.2 |
| Gradient Boosted Trees | 92.2 | 92.1 | 93.7 | 90.4 | 92.3 |
| Support Vector Machine | 91.6 | 91.6 | 93.0 | 89.8 | 91.5 |
| **10 m Accuracy with Additional Points (APs)** | | | | | |
| | Overall Accuracy | Water, User's Accuracy | Water, Producer's Accuracy | Not Water, User's Accuracy | Not Water, Producer's Accuracy |
| Majority Vote Classifier | 91.3 | 94.3 | 89.6 | 87.3 | 93.8 |
| Random Forest | 91.2 | 94.1 | 89.5 | 87.4 | 93.7 |
| Gradient Boosted Trees | 91.1 | 94.0 | 89.4 | 87.3 | 93.7 |
| Support Vector Machine | 91.5 | 94.2 | 89.3 | 87.8 | 92.7 |

Ultimately, while the majority vote classifier offers slightly higher accuracy, it was at the expense of increased computational time required from running the three classifiers. Overall, the results tend to indicate that the type of supervised classifier did not make a major difference with respect to the overall accuracy. The quality, size, and completeness of the training data are likely the most important aspects.

## 4. Discussion

Our results produced similar accuracy levels compared with existing studies, but our approach offers a way to generate surface water maps quickly, on-demand, without manual training data labeling or ancillary training data layers, and evaluated across broad, international geographic regions using a complex and robust accuracy assessment scheme with high error prone points. First, we compared our results to surface water maps produced at the 30m Landsat scale using multiple dates of imagery. In Australia, a study found an overall accuracy of 97% with water producer's and user's accuracies of 93% and 92% respectively [21], as compared with our maximum overall accuracy of 94.3% with user's and producer's accuracies of 97.4% and 91.5% at 20 m across a broader, international geographic area. Two global surface water studies with more comprehensive analysis produced similar accuracies [1,5]; the former had user's and producer's accuracies of 95.2% and 90.3%, and the latter with 93.7% and 96.0% at a monthly time scale. In [56], Dynamic Surface Water Extent (DSWE) layers were used to automatically train a random forest to predict surface water extent in three North American test sites with a median root mean square error of 0.19 across gage sites. The previously mentioned studies produced comprehensive analysis and products with robust accuracy. One study that prototyped the fusion of Sentinel-2 and Sentinel-1 found high 99% overall accuracy, but was limited to a single region [35]. The accuracy would likely be reduced after application to a more international study area; however, the promising result suggests it is likely more accurate than Sentinel-2 only, but at the expense of higher data intensity and manually intensive training methods. Whereas, our methods achieve similar accuracy with fewer inputs and without manual training data creation for easy transferability to analysts and quicker map production. In [29], MNDWI (20 m) and NDWI (10 m) were used across two urban sites with producer's accuracies of 87.3% (20 m) and 88.0% (10 m) and user's accuracies of 88.4% and 90.8% at the most accurate Yantai site. This result is similar to our 20 m index ensemble-only method that was applied across international sites. In [38], OpenStreetMap was used for automated mapping of surface water with Sentinel-2 imagery, which ultimately found

high producer's and user's accuracies ranging from 84% to 99% and 94% to 99% respectively across several test sites in China. This suggests that the automated training data produced from this method is accurate. However, in contrast with that study, our study reported slightly lower accuracy, but did not require OpenStreetMap, which can be a prohibitively large dataset for even modestly sized study areas. Additionally, OpenStreetMap does not have a temporal component, and may not match the imagery which varies by season and year.

Most importantly, because our accuracy assessment included equalized random sampling based on NDWI intervals, it is likely that this method captured more errors and is more robust than a standard simple random or stratified random sampling approach found in all of the previously mentioned studies. Therefore, the different accuracies reported from the mentioned studies are not directly comparable to this study. Moreover, in contrast to our study, many of the mentioned studies cover one or two test sites, or one country. Regarding accuracy, it is also important to note that surface water features smaller than the 20 m or 10 m pixel size would not be detected, and this could play a minor role in accuracy assessment. Recent comprehensive reviews of the results and accuracies of other studies are available [57,58].

In this study, we selected the NDWI, MNDWI, and AWEIsh for our automated training data framework. For the 10 m dataset, we were limited to the NDWI due to the lack of SWIR bands. For the 20 m dataset, we selected these indexes because they are the most commonly employed water indexes. However, we note that other robust water indexes have been developed in recent years that could likely be included or substituted such as the New Water Index (NWI), Tasseled Cap Wetness Index (TCWI), Land Surface Water Index (LSWI), and several others as described in [59]. More recently, the Sentinel-2 water index also appears to produce higher accuracy than the NDWI due to the use of the red-edge and SWIR bands [25]. Similarly, the Weighted Normalized Difference Water Index (WNDWI) also produced high accuracy in difficult-to-map areas, including terrain shadow and urban environments [24].

Previous studies have successfully delineated surface water in glacial and river basin regions using automatically selected thresholds for the spectral indexes based on the Otsu histogram method [60–63]. However, this method is challenging to apply to surface water located near or within built-up, urban environments due to confusion between the two features [29]. Therefore, in our study we determined the optimal thresholds for the spectral indexes by iterating a range of spectral index and SWIR/NIR band thresholds for 12 study sites, and evaluating against the ground-truth points.

The 10 m and 20 m frameworks proposed in this study are easy to implement and require minimal computational capabilities and no additional datasets other than the Sentinel-2 imagery in order to automatically generate the training data and create the surface water maps (a DEM is optional for post-processing and recommended in mountainous terrain). Therefore, we suggest that these frameworks would be ideal for transition into decision support tools for end-users to quickly generate surface water maps for a given study area or date, without the need to obtain a large global dataset, ancillary datasets, or manually create training data.

## 5. Conclusions

This study developed methodologies using Sentinel-2 20 m and 10 m BoA cloud-masked imagery and a method to automatically generate training data for subsequent classification of surface water extent using a random forest classifier. The 20 m method uses an ensemble of three spectral indexes (NDWI, MNDWI, AWEIsh) with optimal index thresholds calculated from iterating a range of index values. This is combined with an optimal threshold applied to the SWIR band for removing false positive water pixels. The 10 m methodology is similar, but only uses NDWI and a NIR band threshold due to band limitations. The accuracy was computed using equalized random points with additional, high error-prone ground-truth points for robust comparison of the thresholds.

Ultimately, after evaluating our comprehensive international ground-truth dataset and iterating through the possible index values, we found the optimal spectral index thresholds of −0.01 (10 m NDWI only) and −0.03 (20 m ensemble of indexes) combined with NIR or SWIR thresholds of 800. A digital elevation model was used to post-process the output over one mountainous study site (T11UQQ) and resulted in an optimal slope threshold of 17 degrees and improved the overall accuracy of that granule/scene by about 21%. We found that the 20 m ensemble method with random forest classifier produced about 4.8% higher overall accuracy than the 20 m ensemble-only method. The overall accuracy for the 20 m ensemble method with automatic training data and a random forest classifier was 94.3% (±1.3%) as compared with 93.4% (±1.5%) for the 10 m method and random forest classifier. The two automated methodologies (10 m and 20 m) were applied across the 12 study sites and resulted in similar total mapped areas of surface water, with an average percent difference of 4.3%. Both methods were effective and accurate, but analysis of the granule-level results suggested that the 10 m framework struggled on highly turbid water bodies (e.g., granule T20LLR), and it had more building and building shadow commission errors (e.g., granule T31UFU). We compared our results with the Sentinel-2 20 m L2A SCL water mask. The SCL water mask had 84.2% accuracy, which was 10.1% lower than our automated method with random forest, and about 5% lower than the ensemble spectral index method, suggesting users should avoid using the water mask that is provided with all Sentinel-2 L2A imagery. Our comparison of the support vector machine, random forest, and gradient boosted trees in conjunction with the training data frameworks, indicated that classifier type did not impact the overall accuracy by more than 1% (with one granule exception) and is likely less important than having a robust training dataset.

Ultimately, this automated training data and surface water map generation method presented here is ideal for development into a decision support tool for end-users who require surface water maps with quick turnaround, using low data intensity, minimal manual input, and robust performance over a wide variety of international landscapes.

**Author Contributions:** Conceptualization, K.L.; Methodology and Formal Analysis, K.L.; Validation, K.L., M.C.M., S.J.B., A.W.H.G. and S.L.L.; Writing–original draft preparation, K.L.; Writing–review and editing, K.L., M.C.M., S.J.B., A.W.H.G. and S.P.G.; Visualization, K.L.; Supervision, K.L., S.J.B. and S.P.G. All authors have read and agreed to the published version of the manuscript.

**Funding:** This research was funded by the U.S. Army Corps of Engineers, Engineer Research and Development Center (ERDC), Geospatial Research business area. Permission to publish was granted by the ERDC Public Affairs Office. Any opinions expressed in this paper are those of the authors, and are not to be construed as official positions of the funding agency.

**Data Availability Statement:** The study used free, publicly accessible data including Sentinel-2 L2A Bottom of Atmosphere reflectance imagery acquired from the European Space Agency's Open Access Hub, and elevation data acquired from the USGS EarthExplorer website.

**Acknowledgments:** The authors thank Nicole Wayant and Jean Nelson for funding procurement and project management, and Elena Sava for scientific discussion related to the topic. The authors thank the four anonymous reviewers for their comments and suggestions.

**Conflicts of Interest:** The authors declare no conflict of interest. The funders had no role in the design of the study; in the collection, analyses, or interpretation of data; in the writing of the manuscript, or in the decision to publish the results.

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
