# Peer review of "Automated Training Data Generation from Spectral Indexes for Mapping Surface Water Extent with Sentinel-2 Satellite Imagery at 10 m and 20 m Resolutions"

_remotesensing, doi:10.3390/rs13224531_

Round 1

Reviewer 1 Report

In this study, the authors proposed an automated method to produce training data for surface water body mapping using 10 and 20 m Sentenl-2 imagery. Overall, this study was written well and suitable to publish in RS. However, this manuscript need further improve before publication.

Comments:

- 10m to 10 m, and 20m to 20 m, a space is necessary

-. The reference type is numbered for RS, this need to be corrected for main text.

-. “…have been experiencing declines in volume and surface water extent over recent decades across various parts of the world” I suggest to add “with an exception for the Tibetan Plateau (Zhang et al., 2020, doi: 10.1016/j.earscirev.2020.103269; Pekel et al., 2016, doi: 10.1038/nature20584)” as the area/level and volume over the Tibetan Plateau have shown an increase.

- “Normalized Difference Water Index (NDWI) or Modified Normalized Difference Water Index (MNDWI), and digital elevation models (DEM)” too many papers are cited once, please decrease to less than five at one place.

- SRTM DEM and Sentinel-1 and Stentinel-2 are combined for water body mapping. However, the spatial resolution for these data is different, how these data are combined?

- The titles for Tables and Figures should be above them.

- Figure 1: The extracted water body boundaries with a red color could be overlapped in Figure 1?

- “A starting threshold of 0 was initially applied…” I don’t think the threshold of 0 is best. How about a dynamic optimal threshold by Otsu method (such as Li et al., 2011, doi:10.1080/01431161.2012.657370 

- How the validation was used for accuracy evaluation?

- Some figures are need to improve such as fronts of Latitude and Longitude in Figure 8 are too busy.

- “%” in Table 3 could be moved to caption.

Reviewer 2 Report

SUMMARY

The paper addresses the research area related to mapping surface water with Sentinel-2 images.

It aims to assess an automated methodology to generate training data for surface water mapping from a single Sentinel-2 granule at 10m (4 bands, VIS/NIR) or 20m (9 bands VIS/NIR/SWIR) resolution without the need for ancillary training data layers.

The author claim that the method presented is ideal for development into a decision support tool for end-users who require surface water maps with quick turnaround, using low data intensity, minimal manual input, and robust performance over a wide variety of international landscapes

As a general comment, the manuscript is fluent and well structured.

MINOR COMMENTs

L130-L140 Please, consider inserting references related to the sentinel-2 details and characteristics.

L188 ‘A starting threshold 186 of 0 was initially applied to each spectral index to separate surface water and non-water 187 pixels, as suggested in the literature’ Please, insert references.

L525 Please, consider clarifying why the scene classification (SCL) maps have not been considered as references in this study.

https://sentinels.copernicus.eu/web/sentinel/technical-guides/sentinel-2-msi/level-2a/algorithm

Reviewer 3 Report

The use of twelve study sites which differ a lot, is one of the positive points of this article. In this sense, authors introduce this and the difficulties presented are very clear described. The introduction, in my opinion, is accurate and clear in this sense.

The methodology proposed seems adequate and it is explained in a way that it can be reproduced by other researchers. It will be of interest if these methods can be used with other sensors/satellite programs. However, it would be of interest to know with detail the software used by authors. ArcGIS Pro is one of them as it is indicated, but a small section of the text indicating the software used (and the way it is used) would be of interest.

The spatial resolution of the pixels is a problem when small water bodies should be detected. It is relevant to recognize this by authors in the results (accuracy assessment and analysis of errors) when they indicate that the 10m product unsurprisingly does a better job at capturing small canals. In fact, as a general rule, researchers should consider the dimension of the pixels and the size of the water bodies to select the adequate sensor and know the inherent error that we can make when using satellite data to determine water surfaces extension.

Please, in order to improve the article, follow the guide for the authors and the style of the journal. Check the following as suggestions:

References in the text preferably indicated by numbers and following the order of appearance in the text (see the guide and style of the journal)

The quality of the figures can be improved. I know that sometimes it is limited because of the output of the programs used, but this can be done with auxiliary software.

Please follow the style of the journal regarding the table headers and the figure captions.

Line 39, check the position of the hyphen at the end to facilitate the reading “distrib-uted”

Line 42, check the position of the hyphen at the end to facilitate the reading “vol-ume”

Line 62, check the position of the hyphen at the end to facilitate the reading “Normal-ized

Line 109, check the position of the hyphen at the end to facilitate the reading “gran-ule”

Line 122, check this “spatial_resolution”

Line 126, check the position of the hyphen at the end to facilitate the reading “man-agers”

Line 143, check the position of the hyphen at the end to facilitate the reading “Can-ada”

Line 183, check the position of the hyphen at the end to facilitate the reading “anal-ysis”

Line 195, check the position of the hyphen at the end to facilitate the reading “gen-erated”

Line 274, check the position of the hyphen at the end to facilitate the reading “thresh-olds”

Line 340, check the position of the hyphen at the end to facilitate the reading “gran-ules/scenes”

Line 358, check the position of the hyphen at the end to facilitate the reading “av-eraging”

Line 363, check the position of the hyphen at the end to facilitate the reading “thresh-old”

Line 364, check the position of the hyphen at the end to facilitate the reading “de-scribed”

Line 392, check the position of the hyphen at the end to facilitate the reading “thresh-old”

Line 466, check this please “index-“, I think it is not necessary the use of an hyphen

Line 522, check the position of the hyphen at the end to facilitate the reading “man-ual”

Line 572, check the position of the hyphen at the end to facilitate the reading “Normal-ized

Reviewer 4 Report

The authors have presented a significant amount of work in their manuscript titled "Automated training data generation from spectral indexes for 2 mapping surface water extent with Sentinel-2 satellite imagery 3 at 10m and 20m resolutions". Since authors mentioned about decision support tools - which is a significant applications of such tools, it may help if authors can provide some details on developing decision support tools.

Round 2

Reviewer 1 Report

The authors have addressed all my comments, and I agree with the publication of this manuscript. 

Author Response

We thank the reviewer for their comments and support.

Reviewer 2 Report

SUMMARY

The paper has been improved since the first review round.

MINOR COMMENTs

Figure 6-7-8 Please consider inserting the SCL maps in order to show the differences.

Author Response

Point 1: Figure 6-7-8 Please consider inserting the SCL maps in order to show the differences.

Response 1: We thank the reviewer for this suggestion. We modified figures 7, 8, and 10 to include the SCL maps. We were unable to do the same for figure 6 as that figure became overcrowded and too low resolution to differentiate the map differences.

Reviewer 3 Report

Thanks for improving the article following recomendations.

Author Response

(The authors gave the same response as above.)
